# Effect of Post-Weld Heat Treatment on Microstructure and Fracture Toughness of X80 Pipeline Steel Welded Joint

**DOI:** 10.3390/ma15196646

**Published:** 2022-09-25

**Authors:** Xueli Wang, Dongpo Wang, Lianshuang Dai, Caiyan Deng, Chengning Li, Yanjun Wang, Ke Shen

**Affiliations:** 1Key Laboratory of Advanced Joining Technology of Tianjin, Department of Materials Science and Engineering, Tianjin University, Road Weijin 92, Tianjin 300072, China; 2Pipe China North Pipeline Company, Road Xinkai 408, Langfang 065000, China; 3China Oil & Gas Piping Network Corporation, Beijing 100013, China; 4Pipe China Beijing Pipeline Co., Ltd., Beijing 100101, China

**Keywords:** pipeline steel, coarse-grained heat-affected zone, crack tip opening displacement

## Abstract

In the current study, post-weld heat treatment (PWHT 580 °C) was used for an X80 pipeline steel-welded joint, and the fracture toughness of the welded joint was investigated using a crack tip opening displacement (CTOD) test. The relationship between microstructure evolution and fracture toughness is also discussed in this study. The results showed that the weld center mainly consisted of acicular ferrite (AF). The subcritical heat-affected zone (SCHAZ) consisted of a large amount of fine polygonal ferrite and some AF, and it maintained the rolling state of the base metal. The microstructure of the coarse-grained heat-affected zone (CGHAZ) was composed of granular bainite (GB) and M/A constituents, the latter of which decreased after the PWHT. The CTOD values of the weld center were in the range of 0.18–0.27 mm, while those of the CGHAZ were in the range of 0.02–0.65 mm. A brittle fracture occurred in the CGHAZ for both the as-welded and PWHT samples; the CTOD values were 0.042 mm and 0.026 mm, respectively. The CTOD values of the SCHAZ’s location were in the range of 0.8–0.9 mm. The PWHT did not deteriorate the microstructure of the CGHAZ and had little influence on the fracture toughness of the X80 pipeline steel-welded joint; it ensured the fracture toughness of the welded joints and reduced the welding residual stress.

## 1. Introduction

Oil and gas transmission pipelines are important facilities used in China, and with the increasing demand for their use spanning greater distances and withstanding higher pressure levels, the safety of long-distance oil and gas transmission pipelines is also subject to scrutiny [1]. In order to improve the efficiency of long-distance oil and gas pipelines, the use of high-grade pipeline steel, such as X70 and X80, has increased [2,3]. In recent years, dozens of X70 and X80 high-grade pipeline steel-welded joint fracture accidents have occurred at home and abroad, resulting in severe injuries and economic losses, thus making the safety of high-grade pipeline steel-welded joints one of the most important issues that need to be addressed urgently, at present, in the oil and gas industry [4,5].

In the welding process of pipeline steel, the heat-affected zone near the fusion line experiences a longer period of high-temperature-welding thermal cycling (generally no less than 1200 °C), resulting in the formation of a coarse-grained heat-affected zone (CGHAZ) [6,7,8]. In addition, the welding thermal cycle inevitably produces high welding residual stresses. Considerable evidence has shown that pipeline steel-welded joints produced severe structural stress (caused, for example, by variable wall thickness and misalignment), exhibited insufficient strength regarding welded metal and the GCHAZ, and considerable welding residual stresses, which are all important factors that cause the failure of high-grade pipeline steel-welded joints, affecting the entire safety structure of the welded joint [9,10].

For pipeline steel-welded joints, post-weld heat treatment in the range of 100–200 °C below the A_C1_ temperature is used to reduce the residual stress of general steel according to the GB/T 16923-2008 “Normalizing and Annealing of Steel Parts” standard. Utilizing the heat input from adjacent weld passes and subsequent weld layers to provide tempering for the base or weld metals, namely, temper bead welding, was also used in situations where the PWHT was time consuming, expensive, or impractical [11]. Several researchers have proven that tempering performed between 500–650 °C could relieve residual tensile stress for general steel [12,13]. Residual stress and hardness were decreased by increasing the heat-treatment temperature and holding time [14,15,16]. An annealing temperature that is too high for pipeline steel can affect the microstructure of the welded joint, while residual stress cannot be effectively reduced at a temperature that is too low [17,18]. Residual stress reduction can significantly reduce the deterioration rate of the fracture toughness of the pipeline steel-welded joint [12,19,20]. However, post-weld heat treatment may exhibit further effects on the microstructure and fracture toughness of the welded joint. Heat treatment below A_c1_ causes the tempering of martensite and ensures the sufficient volume fraction of retained austenite, resulting in the toughness of the 410NiMo welded metal [21]. Xu et al. investigated the effect of stress-relief annealing on the fatigue properties of X80 welded pipes [22], and the results show that stress-relief annealing has a greater recovery effect on high-energy dislocations, and the high-energy dislocation motion resulted in a change in the fatigue properties.

The use of post-weld heat treatments to eliminate residual stress is a commonly used process, but its effect on microstructure and fracture toughness have not been revealed. In the current paper, the effects of post-weld heat treatment on the microstructure evolution of an X80 pipeline steel-welded joint at high temperatures is investigated to explore whether post-weld heat treatment affects the microstructure and fracture toughness of the welded joint and provides a basis for understanding the X80 pipeline post-weld heat-treatment process and performance evaluation.

## 2. Experimental Procedures

In the current study, diameters of 1422 mm for the X80 pipeline steels were welded, including a wall thickness of 21 mm and length of 1000 mm. For the welding process, we employed a gas metal arc welding (GMAW) process with an automatic pipeline welder, A610 Argon arc welder, with shielding gas values of 80% Ar and 20% CO_2_. Lincoln Pipeliner 80Ni1 welding wire with a diameter of 1.0 mm was used in the welding process. There are 6 layers of the welded joint including backing weld, hot welding, filling welding layer, and cosmetic welding. Table 1 shows the details of the welding parameter for each weld layer. The chemical compositions of X80 were tested with a GS1000-11 optical emission spectrometer (from OBLF of Witten, Germany), and Lincoln Pipeliner 80Ni1 welding wire chemical compositions obtained by the manufacturer. Their chemical compositions are presented in Table 2; the yield strength range was 550–700 MPa, and the tensile strength range was 620–820 MPa for the X80 pipeline steel used in this study. The welded joint was heated to 580 °C at a rate of 3 °C/min and was held for 1 h, and was then cooled to room temperature in the air. 

The CTOD test was conducted according to the BSENISO15653 (2018) and DNVGL-ST-F101 (2017) standards to obtain the CTOD value of the weld center, the CGHAZ (the fusion line), and the SCHAZ (5 mm from the fusion line) for the AW and PWHT conditions. The sampling locations and macroscopic cross-section morphology of the welded joint are presented in Figure 1. B × 2B (B = 17 mm)-type specimens with a length of 120 mm (≥4.6 W) were used. The notch direction of the weld center and heat-affected zones (the CGHAZ and SCHAZ) were NP and NQ, respectively, where N was the vertical weld direction, P was the direction of the parallel weld, and Q was the direction of the weld thickness. The metallographic samples were polished and etched using a 4% nitric acid alcohol solution. A microstructure analysis was performed using a Smartzoom5 Zeiss super-field microscope, an OLYMPUS GX51 optical microscope (from OLYMPUS of Tokyo, Japan), and a JSM-7800F thermal field emission scanning electron microscope (from JEOL of Tokyo, Japan). The metallographic test was conducted according to the GB/T13298-2015 inspection methods of microstructures for metals. A microhardness test was performed on a 432SVD automatic turret-type Vickers hardness tester (from Shanghai HDNS Precision Instruments Co., Ltd. of Shanghai, China) with a loading load of 10 kgf and a dwell time of 15 s.

## 3. Results and Discussions

### 3.1. Effects of Post-Weld Heat Treatment on Fracture Toughness

CTOD loading curves and the characteristic values of the welded joint for the as-welded and post-weld heat-treated conditions at −20 °C are presented in Figure 2. The CTOD value of the weld center was in the range of 0.18–0.27 mm, while that of the CGHAZ was in the range of 0.02–0.65 mm. There was one sample for the CGHAZ that exhibited brittle fractures in both the welded and PWHT states; the CTOD values of the two samples were 0.042 mm and 0.026 mm, respectively. The CTOD value of the SCHAZ was in the range of 0.8–0.9 mm.

It can be observed from the CTOD test results that the stress-relief heat treatment did not improve the fracture toughness of the X80 pipeline steel-welded joint whilst, simultaneously, having no adverse effects. The heat-treatment process at 580 °C did not affect the problem of brittle fracture occurring in the CGHAZ, and the risks of brittle fracture after stress-relief heat treatment were still encountered. The fracture toughness properties of the weld center and the SCHAZ were more stable; no brittle fracture phenomenon occurred. The CTOD value of the weld center was relatively low, where the CTOD values of the CGHAZ and SCHAZ were 3–4 times that of the weld center.

### 3.2. The Effect of Post-Weld Heat Treatment on Microstructure Evolution

The microstructure of the X80 pipeline steel was mainly composed of polygonal ferrite (PF) and some AF. For the two conditions investigated in this study, the metallographic photographs of the welded joint at different locations are presented in Figure 3. The weld center was dominated by large, columnar grains, which were composed of staggered AF, some grain boundary ferrite (GBF), and a small amount of bainite. The fine-grained heat-affected zone (FGHAZ) mainly comprised GB, some lath bainite (LB), and M/A constituents [23]. The microstructure of the SCHAZ was similar to that of the base material. However, it was subjected to the effects of heat input during the welding process and underwent a process similar to tempering; the microstructure maintained both PF and AF components.

The microstructure of the post-weld heat-treated samples was similar to that of the as-welded samples in different locations of the welded joint, which mainly exhibited changes in grain size and the recovery of the substructure. The amount of GBF in the weld center was increased after post-weld heat treatment, while that of M/A constituents and GB in the FGHAZ was also reduced, the sizes of M/A constituents and GB were reduced further, and the sizes of AF and PF in the SCHAZ became finer.

The CGHAZ directly affected the overall performance of the welded joint, and its microstructure is presented in Figure 4a,b. The as-welded sample was mainly composed of GB and LB, and some M/A constituents. The CGHAZ was still dominated by GB and LB after post-weld heat treatment.

The statistical results obtained for the grain sizes at different locations of the welded joints are presented in Figure 5a. It is noteworthy that the post-weld heat treatment only resulted in the slight increase in the grain size in the FGHAZ, and the remaining regions presented a decrease in the grain size. The grain size in the FGHAZ was 3.66 μm, obtained from the welded joint, and was increased to 4.74 μm after heat treatment. It is generally assumed that the grain size will increase after the PWHT, but for pipeline steel, such as thick-wall carbon steel pipes, Rajamurugan et al. reported that the microstructure was enriched from coarse grain to fine grain after the local PWHT process (maximum temperature ranged from 500 °C to 800 °C) [12,13,24]. In the X80 joint, the grain size varied in a small range, which was less than two microns, and it exhibited no significant increase or decrease after the PWHT. As previously mentioned, the PWHT using 580 °C mainly reduced the residual stress and had little effect on the grain size. The microstructure of a welded joint is not uniform [25], and it should be pointed out that the welded sample and the heat-treated sample were taken from different positions of the same girth weld. There were slight differences in the microstructure of the different sections in the weld during the welding process. These results led to slight changes in the grain size test results. However, in this study, the grain size did not significantly increase or decrease after the PWHT.

The heat-treatment process applied in the current experiment mainly affected the substructure and distribution of low-angle grain boundaries. The change to a low-angle grain boundary (1–5°) was related to the state of the substructure, such as dislocations [12,26,27]. The increase in the low-angle grain boundaries contributed to the improvement of the strength and toughness of the material [28]. As shown in Figure 5b,c, the grain-orientation-angle fraction distribution of 1–15° increased, while the 15–180° angle decreased in the WZ, CGHAZ, and FGHAZ after the PWHT. In the SCHAZ, the 1–15° grain-orientation-angle fraction decreased and the 15–180° angle increased after the PWHT. 

Based on the results obtained from these microstructural analyses, the heat-treatment process conducted at 580 °C for one hour resulted in a partial recovery of the microstructure; both the grain size and grain-orientation angle increased or decreased at a minor range. Although the grain size in the FGHAZ and grain-orientation angle in the SCHAZ presented different patterns of change after the PWHT, there was no significant change in the CTOD properties in the CGHAZ because the heat treatment did not deteriorate the microstructure.

The microstructure evolution affected the microhardness of the welded joints, especially in the heat-affected zone. The microhardness distributions of the welded joints for the two conditions we investigated are presented in Figure 6. Compared to the as-welded samples, the microstructure of the PWHT samples recovered. The microhardness of the weld center and CGHAZ was basically unchanged due to the relatively stable AF and PF. The microhardness of the heat-affected zone significantly reduced, which was attributed to prominent inhomogeneity and the recovery of the microstructure.

### 3.3. Crack Propagation and Fractography

The EBSD maps of the secondary crack close to the main crack extension tip of the CTOD samples are presented in Figure 7, where the samples for the two conditions exhibited the same characteristics since the post-weld heat treatment did not significantly change the microstructure of the welded joint. It was thought that the brittle fracture presented in the CGHAZ was a complex combination of factors, such as a precipitated phase and the coarse-grain size; the PWHT conducted at 580 °C could only contribute to the recovery of the microstructure to decrease the residual stress. The brittle fracture occurring in the CGHAZ was difficult to eliminate by heat treatment.

As shown in Figure 7a,b, the GBF always existed in the WZ after the PWHT. GBF as a brittle phase has a considerable effect on the low temperature toughness of steels. The fine AF in the WZ contributed to preventing crack propagation, but the GBF easily provided a path for crack growth during the CTOD test. In the CGHAZ, the prior austenite grain boundary (PAGB) could be detected in the secondary crack tip, and the cracks penetrated the LB in Figure 7b,e. It can be observed in Figure 7c,f that the crack followed the rolling direction in both the welded and heat-treated joints. 

The grain-orientation angle distributions presented no significant differences for the two conditions, since the PWHT did not change the microstructure of the welded joint (Figure 8). The plastic strain distribution in the post-weld heat-treated samples was highly localized around the secondary cracks, where it was greatest immediately adjacent to the crack edges, gradually decreasing with an increase in the distance from the cracks (Figure 9d,e). Conversely, the plastic strain distribution in the as-welded samples was more diffuse around the crack (Figure 9a–c). The recovery of the substructure for the post-weld heat-treated samples was expected to cause highly localized deformation patterns, while the dislocation angles caused the slip activity to be less localized in the as-welded samples, resulting in the deformation patterns being more diffuse.

The fracture surfaces of the CTOD samples are presented in Figure 10. The macroscopic fracture morphology can be divided into mechanical gap, pre-crack, and fracture zones (Figure 10a). Figure 10b presents the morphology obtained from the brittle fracture sample in the CGHAZ in a welded and PWHT state. It can be observed that the CTOD crack-propagation zone was close to the pre-crack’s tip. The CTOD crack-propagation zone is related to the toughness, and the better the toughness, the larger the CTOD crack propagation zone [29]. Dimples were detected in the CTOD crack-propagation zone, and the brittle fracture samples had a narrow crack-propagation region. The other samples without brittle fractures presented a large CTOD crack-propagation zone compared to the brittle fracture samples, as presented in Figure 10c.

The reason for the occurrence of brittle fractures was mainly related to the coarsening of the CGHAZ. The PWHT can reduce residual stress, but cannot improve the microstructure of the coarse-grain zone, so the brittle fracture in the CGHAZ still exists after the PWHT. On the other hand, the inhomogeneous distributions of the GB and M/A constituents simultaneously led to the occurrence of localized vulnerable areas, which triggered the brittle fracture occurring in the CGHAZ [23,30].

## 4. Conclusions

(1) The weld center of the X80 pipeline steel-welded joint mainly consisted of AF. The SCHAZ was composed of a high amount of fine polygonal ferrite (PF), and it still maintained the rolling direction of the base material. The microstructure of the CGHAZ was composed of GB and some M/A constituents. The stress-relief heat treatment conducted at 580 °C for 1 h did not change the microstructure of the welded joint, and no deterioration of the welded joint occurred.

(2) The stress-relief heat treatment had little effect on the fracture toughness of the welded joint. The CTOD values of the weld center were in the range of 0.18–0.27 mm, while those of the CGHAZ were in the range of 0.02–0.65 mm. The SCHAZ exhibited the highest CTOD value over the range of 0.8–0.9 mm. Therefore, it could reduce the residual stress and ensure the fracture toughness of the welded joint.

(3) Brittle fracture occurred in both the CGHAZ for the as-welded and PWHT samples; the CTOD values were 0.042 mm and 0.026 mm, respectively. The coarsening of the grain size on the fusion line was the main reason for the occurrence of brittle fractures. In addition, the non-uniform distributions of M/A constituents led to the occurrence of localized vulnerable areas, which triggered the brittle fractures occurring in the CGHAZ.

## Figures and Tables

**Figure 1 materials-15-06646-f001:**
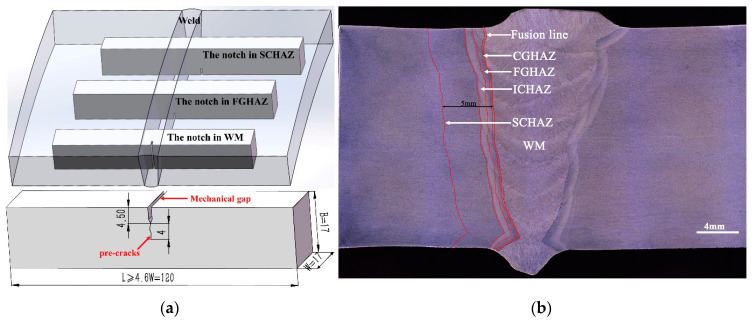
Sampling location of CTOD and macroscopic photograph of the welded joint: (**a**) sampling location of CTOD; (**b**) macroscopic photograph of the welded joint.

**Figure 2 materials-15-06646-f002:**
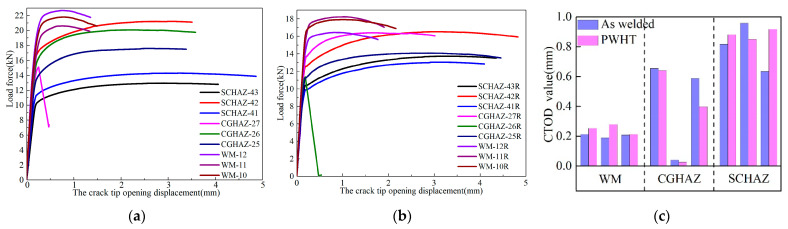
CTOD loading curve and statistical results of characteristic values: (**a**) CTOD loading curve of welded sample; (**b**) CTOD loading curve of PWHT sample; and (**c**) CTOD characteristic value statistics.

**Figure 3 materials-15-06646-f003:**
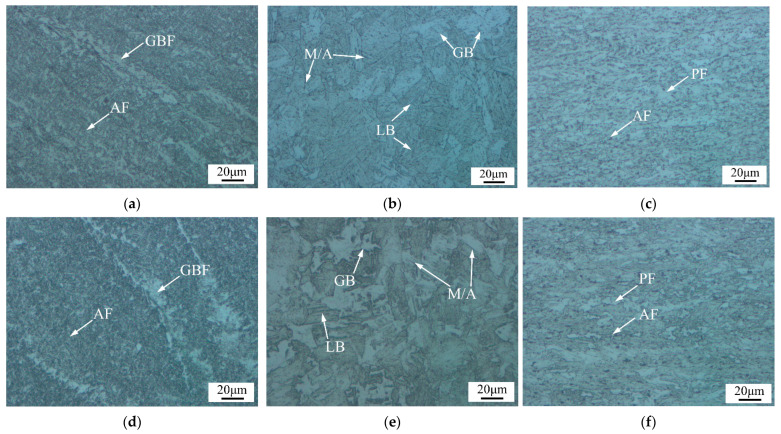
The microstructure obtained from the welded and heat-treated joints: (**a**) the center of the welded joint; (**b**) FGHAZ of welded joint; (**c**) SCHAZ of welded joint; (**d**) the center of the heat-treated joint; (**e**) FGHAZ of heat-treated joint; and (**f**) SCHAZ of heat-treated joint.

**Figure 4 materials-15-06646-f004:**
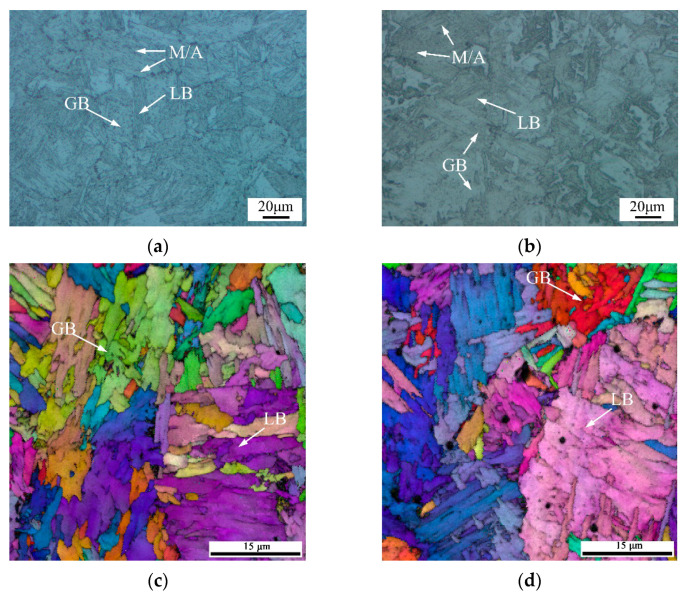
Microstructure of CGHAZ: (**a**) CGHAZ of welded joint; (**b**) CGHAZ after heat treatment; (**c**) EBSD of CGHAZ of welded joint; (**d**) EBSD of CGHAZ of heat-treated joint.

**Figure 5 materials-15-06646-f005:**
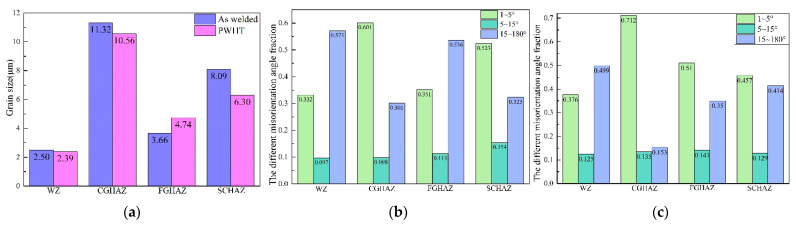
(**a**) Grain size following PWHT; (**b**) the grain-orientation angle of the welded joint; and (**c**) the grain-orientation angle after PWHT.

**Figure 6 materials-15-06646-f006:**
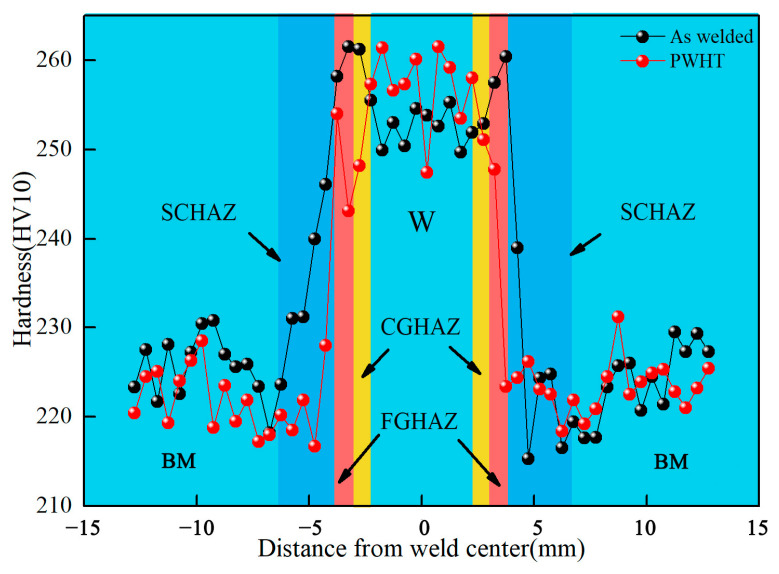
Hardness distribution of the welded and heat-treated joints.

**Figure 7 materials-15-06646-f007:**
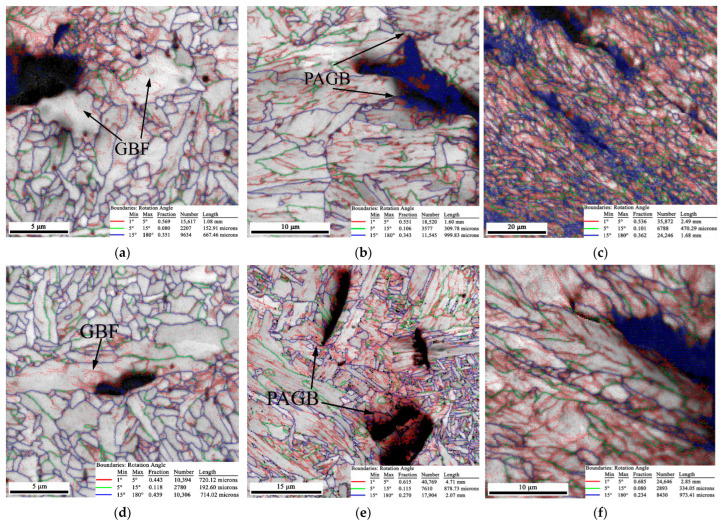
The EBSD map of the secondary crack obtained from the welded sample: (**a**) WZ; (**b**) CGHAZ; and (**c**) SCHAZ, and PWHT samples: (**d**) WZ; (**e**) CGHAZ; and (**f**) SCHAZ.

**Figure 8 materials-15-06646-f008:**
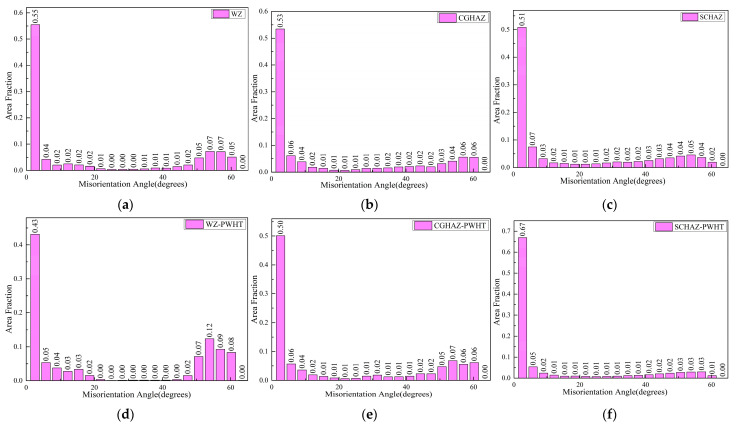
The grain orientation angles obtained from the welded sample: (**a**) WZ; (**b**) CGHAZ; and (**c**) SCHAZ, and PWHT sample: (**d**) WZ; (**e**) CGHAZ; and (**f**) SCHAZ.

**Figure 9 materials-15-06646-f009:**
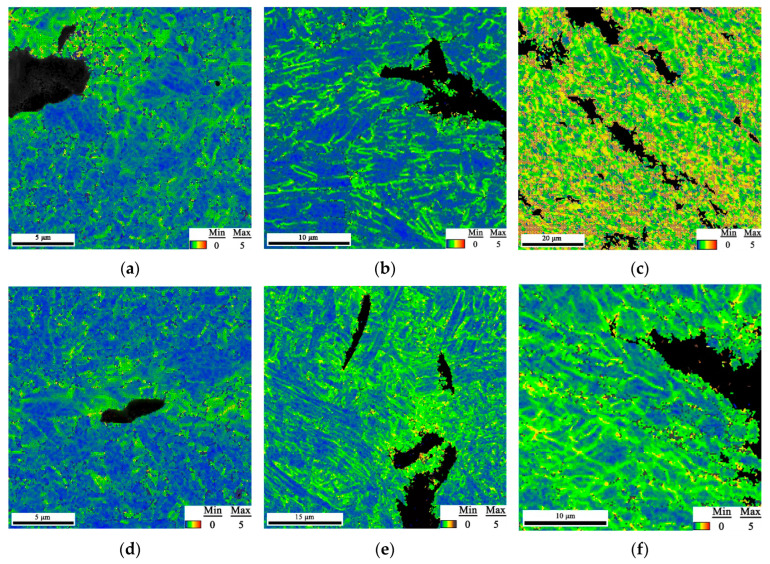
The KAM map of the secondary crack obtained from the welded sample: (**a**) WZ; (**b**) CGHAZ; and (**c**) SCHAZ, and PWHT sample: (**d**) WZ; (**e**) CGHAZ; and (**f**) SCHAZ.

**Figure 10 materials-15-06646-f010:**
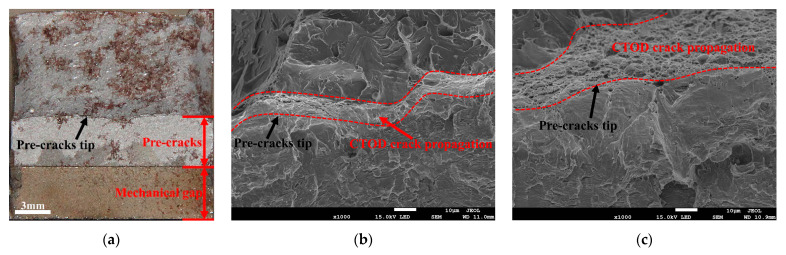
The fracture surface of the CTOD sample: (**a**) macroscopic; (**b**) brittle sample; and (**c**) non-brittle sample.

**Table 1 materials-15-06646-t001:** The details of welding parameters for each weld layer of X80 joint.

Layer	Bead	Welding Voltage (V)	Welding Current (A)	Welding Speed (cm/min)	Shield Gas Flow Rate (L/min)
Backing weld-1	1-1	20–21	180–200	50–60	27–30
Hot welding-2	2-1	21–22	160–180	70–80	
Filling welding-3	3-1	22–24	100–180	40–50	
	3-2				
Filling welding-4	4-1				
	4-2				
Filling welding-5	5-1				
	5-2				
Cosmetic welding-6	6-1	24–25	90–110	50–60	
	6-2				
	6-3				

**Table 2 materials-15-06646-t002:** The chemical composition of X80 pipeline steel and welding wire.

	Chemical Composition (wt%)										
X80 pipeline steel	C	Si	Mn	Cr	Mo	Ni	Nb	Ti	S	V	-
	0.042	0.200	1.830	0.330	0.005	0.160	0.090	0.012	0.002	0.005	-
Welding wire (80Ni1)	C	Si	Mn	Cr	Mo	Ni	Cu	Ti	Al	P	S
	0.089	0.680	1.540	0.030	0.008	0.940	0.022	0.067	0.006	0.006	0.005

## Data Availability

Not applicable.

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
