# Peer review of "Effect of Post-Weld Heat Treatment on Microstructure and Fracture Toughness of X80 Pipeline Steel Welded Joint"

_materials, 2022, doi:10.3390/ma15196646_

Round 1

Reviewer 1 Report

Dear Authors,

I have read your paper titled: "Effect of post-weld heat treatment on microstructure and fracture toughness of X80 pipeline steel welded joint".

The topic is important and requires deep investigations. Paper fulfills the aims and scope of Materials journal. I have some suggestions, which may help you in improving the text.

General remarks:

  • Please add the quantitative results into the abstract.

Introduction:

  • You have describer traditional post-weld heat treatment well. However, recently other methods providing same effect are widely investigated. In my opinion, it is wortk to mention about them, e.g., "temper bead welding" (https://scholar.google.com/scholar?as_ylo=2018&q=%22temper+bead+welding%22&hl=pl&as_sdt=0,5), which is used to provide local heat treatment of the britte microstructures in the HAZ.

Experimental procedures:

- This section requires serious improvement.

- You have presented the diameter od used matarial. What about other dimensions as wall thickness and length of specimen? Please show this in the paper.

- I cannot find the description of used materials. Please show the chemical composition of base metal and filler material. Moroever, the basic mechanical properties as yield point, tensile strength and elongation have to be presented. These information are crucial for creating full joint without imperfections.

- None information about welding technology is presented. How many stitches (beads) were used? Which welding parameters were selected and why? These are crucial information for your paper.

- Metallographic test is standarized test. Please add information about used standard.

Results and Discussion:

- Results are presnted well. You have analyzed results. However, I cannot find deep scientific discussion. You should compare your results with other scientific papers. It allows to underline the biggest advantages from your investigations. Moreover, it will underline the necessity and novelty of your work.

- Fig. 10a - scale bar is missing.

- Have you observed any welding imperfections in specimen?

Conclusions:

- Please support conclusions with the quantitative results.

Reviewer 2 Report

The manuscript should be rejected. The major reasons are as the following.

1. The quality and level of the manuscript haven't achieved the standard of the Journal. Discussion about the received results and revealed effects is at a low level. In fact, the article is a description and documentation of the experiment.

2. There is little scientific merit or contribution in the manuscript.

Other:

- The authors do not provide any data on the studied steel, except for its strength grade.

- The Method Part is not detailed.

- If the increase in grain size after relatively low-temperature exposure can still be associated with the activation of diffusion, then the decrease in grain size is generally incomprehensible.

- The emphasis in the study is on reducing residual stresses, however, the authors did not measure these stresses.

- Line 61-67. The phrase should be divided into at least 2 parts.

- What does the series of three different curves in Figure 2 and the numbers in their designations mean? 

- The English expression of this manuscript needs some further improvement.

Round 2

Reviewer 1 Report

Dear Authors,

Some issues needs clarification:

1. " The welding voltage and current were 22V 83 and 140A, respectively"

How many layers (beads/stitches) have been performed? Each with the same parmeters? If yes, why the same were used for each layer? Welding technology usualy requires different parmeters for groove stitch and for other stitches.

2. Table 1 - the source of presented values is unknown. Have you tested chemical compositions? If yes, which method was used? If not, please mark the source - stnadard, manufactutrer data or other paper.

Reviewer 2 Report

After revision, the paper has been improved. While a few points can still be improved, they are not fundamental. Therefore, I believe that the paper can be published as it is.

Author Response

We deeply appreciate the time and effort you’ve spent in reviewing our manuscript. Your comments are really thoughtful and helpful for our work.